# Polyoxovanadate-Based Cyclomatrix Polyphosphazene Microspheres as Efficient Heterogeneous Catalysts for the Selective Oxidation and Desulfurization of Sulfides

**DOI:** 10.3390/molecules27238560

**Published:** 2022-12-05

**Authors:** Yinghui Hu, Diping Huang, Jing Yan, Zhiliang Miao, Lize Yu, Ningjing Cai, Quanhai Fang, Qiuyu Zhang, Yi Yan

**Affiliations:** 1Department of Chemistry, School of Chemistry and Chemical Engineering, Key Laboratory of Special Functional and Smart Polymer Materials of Ministry of Industry and Information Technology, Northwestern Polytechnical University, Xi’an 710129, China; 2Queen Mary University of London Engineering School, Northwestern Polytechnical University, Xi’an 710129, China

**Keywords:** polyoxometalate, cyclomatrix polyphosphazene, catalytic oxidation, sulfides

## Abstract

The [V_6_O_13_]^2−^ cluster is successfully immobilized to the polymeric framework of cyclomatrix polyphosphazene via the facile precipitation polymerization between the phenol group symmetrically modified [V_6_O_13_]^2−^ and hexachlorocyclotriphosphazene. The structure of the as-prepared polyoxometalate-containing polyphosphazene (HCCP-V) was characterized by FT-IR, XPS, TGA, BET, as well as SEM and zeta potential. The presence of a rigid polyoxometalate cluster not only supports the porous structure of the polymeric framework but also provides an improved catalytic oxidation property. By using H_2_O_2_ as an oxidant, the as-prepared HCCP-V exhibited improved catalytic oxidation activity toward MPS, DBT, and CEES, which can achieve as high as 99% conversion. More importantly, the immobilization of POMs in the network of cyclomatrix polyphosphazene also provides better recyclability and stability of the heterogeneous catalyst.

## 1. Introduction

Metal-containing polymers (MCPs) receive broad attention due to the synergetic effect from the functionality of the metal unit and the processability of the polymeric framework [1,2,3,4,5]. As one of the potential candidates for the functional metal units [6], polyoxometalates (POMs) have been introduced to the polymeric framework via either covalent modification [7,8] or ionic interactions [9,10], due to their diverse applications in many fields, such as catalysis, energy conversion, memory storage, medicine, and so on [11,12,13]. The variety of POMs structures and components as well as the topology of polymers provide rational design of POMs-containing polymers for specific application [14], especially in the field of flexible electronics [15].

As a special class of POMs, vanadium-containing POMs have received extensive attention in recent years due to their unique properties [16]. The rich redox properties of vanadium enabled its application in the construction of a high-performance zinc-ion battery [17] as well as electrochemical capacitors [18]. Polyoxovanadates (POVs) also showed interesting biomedical applications, as antidiabetic, antibacterial, antiprotozoal, antiviral, and anticancer drugs [19]. For example, both decavanadate and metformin-decavanadate exert antiproliferative effects on melanoma cells at 10 times lower concentrations than monomeric vanadate [20].

More importantly, POVs have also been reported to be powerful catalysts for the catalytic oxidation of sulfides owing to their multiple redox state [21,22]. For example, isopolyoxovanadate [H_3_V_10_O_28_]^3−^ showed high catalytic activity in oxidation of dibenzothiophene (DBT) to corresponding sulfone by using molecular oxygen as an oxidant under mild conditions [23]. Moreover, POVs also exhibited excellent catalytic properties in the decontamination of chemical warfare agents, such as sulfur mustard. For example, Hu and coworkers reported that H_13_[(CH_3_)_4_N]_12_[PNb_12_O_40_(V^V^O)_2_⋅(V^IV^_4_O_12_)_2_]⋅22 H_2_O can effectively catalyze both the hydrolysis of the nerve agent simulant diethyl cyanophosphonate (DECP) and selective oxidation of the mustard simulant 2-chloroethyl ethyl sulfide (CEES) [24]. To improve the recyclability and stability of POVs-based catalysts, different strategies have been developed to immobilize POVs: (i) Introduce POVs to the metal-organic framework (MOF) or other porous materials [25], such as zeolite, via non-covalent interaction. For example, several POVs-based MOFs, [Co(HDTBA)V_2_O_6_] [26], [Co_2_L_0.5_V_4_O_12_]⋅3DMF [27], and [Cu(mIM)_4_]V_2_O_6_ [28], were synthesized and can efficiently catalyze the H_2_O_2_- or *tert*-butyl hydroperoxide-based oxidation of sulfides and oxidative detoxification of the sulfur mustard simulant CEES. (ii) Immobilize POVs to the polymeric framework via covalent bonds [29]. For example, ring-opening metathesis polymerization of a POMs-based norbornene monomer was developed by Wang’s group [30]. Hill and coworkers reported the POMs-based gelating network via the polycondensation between [H_3_V_10_O_28_]^3−^ and polyol precursors [31]. To extend the polymeric framework of POM-based MCPs, we developed a facile preparation method of POMs-containing cyclomatrix polyphosphazenes through the precipitation polymerization of hydroxyl-functional POMs and hexachlorocyclotriphosphazene [14]. It is hypothesized that if POVs can be introduced to such structure, the homogeneous distribution of POVs in the cyclomatrix framework and its porosity may provide multiple interaction sites between the catalyst and the substrate, which will enable better oxidation activity and recyclability for the selective oxidation of sulfides [32,33].

Herein, we report the immobilization of [V_6_O_13_]^2−^ to the cyclomatrix polyphosphazene microspheres via the precipitation polymerization between phenyl symmetrically modified [V_6_O_13_]^2−^ and hexachlorocyclotriphosphazene. The resulted HCCP-V displayed versatile properties in the selective oxidation of different sulfides, including methyl phenyl sulfide (MPS), DBT, and CEES. It is believed that the design of POMs-based cyclomatrix polyphosphazene microspheres may provide a new platform for the construction of POMs-based MCPs toward task-specific applications.

## 2. Results and Discussion

### 2.1. Structural and Morphological Characterization of HCCP-V

As shown in Figure 1A, there are mainly two strategies to incorporate polyoxovanadates (POVs) into the polymeric framework: (i) poly-condensation between [H_3_V_10_O_28_]^3−^ and polyol precursors [31], and (ii) free radical polymerization of vinyl groups symmetrically modified [V_6_O_13_]^2−^ [34,35,36]. Owing to the multiple redox states of vanadium, the yield of such condensation is usually low, limiting the practical application of POV-based materials. To overcome such disadvantage, inspired by our previous study on the polyoxometalate-containing cyclomatrix polyphosphazene microspheres [14], precipitation polymerization was used to immobilize functional POV to the framework of cyclomatrix polyphosphazene. As shown in Figure 1B, the synthetic methodology of target POV-containing cyclomatrix polyphosphazene HCCP-V is very straightforward. To improve the reactivity of the hydroxyl group of V_6_O_13_-OH toward P−Cl, phenol groups were introduced via the 1,1′-carbonyldiimidazole (CDI)-mediated condensation. Then, the target POV-containing cyclomatrix polyphosphazene microsphere can be facile-prepared via the precipitation polymerization between V_6_O_13_-PhOH and hexachlorocyclotriphosphazene (HCCP) with the aid of triethylamine as a base. As shown in Appendix A, the peaks at 9.04, 8.19, 7.17, and 6.67 ppm can be assigned to the phenol groups, indicating the successful grafting of phenol groups to the [V_6_O_13_]^2−^ cluster, which is in good accordance with the literature [14]. Meanwhile, the modification of phenol groups can also be demonstrated by the characteristic peaks of N–H at 3280 cm^−1^ and benzene ring at 1606 and 1553 cm^−1^ (Appendix A). Moreover, the presence of characteristic peaks of V–O and V–O–V stretching at 951, 835, and 712 cm^−1^ indicated the structure integrity of the V_6_O_13_ cluster after CDI modification.

The phenol groups endowed V_6_O_13_-PhOH with improved reactivity towards HCCP, facilitating the preparation of HCCP-V. As shown in Figure 1A, the peaks at 1238 and 906 cm^–1^ can be assigned to P=N and P–O–Ph [37,38], indicating the successful precipitation polymerization. Furthermore, the characteristic peaks of V–O at 955 cm^−1^ and V–O–V at 805 and 706 cm^−1^ indicated that the vanadium clusters were introduced into the crosslink network. The characteristic peaks of the N–H bond and benzene ring were found at 3140, 1543 cm^−1^ and 1508, 1606 cm^−1^, indicating the existence of V_6_O_13_-PhOH. The introduction of the [V_6_O_13_]^2−^ cluster to the framework of HCCP-V can be further proven by corresponding thermal gravimetric analysis (TGA). As shown in Figure 1B, the presence of [V_6_O_13_]^2−^ not only improved the thermal stability of the cyclomatrix polyphosphazene but also increased the residual weight at a high temperature. According to the 42% residual of P_2_O_5_ and V_2_O_5_ at 900 °C, it can be calculated that there is ca. 18.85 wt.% of the [V_6_O_13_]^2−^ cluster in the resulted HCCP-V (calculation is presented in the Appendix A).

To further characterize the detailed structure of HCCP-V, X-ray photoelectron spectroscopy (XPS) was used. As shown in Figure 2A,B, the successful introduction of V_6_O_13_-PhOH to the polymeric framework can be proven by the presence of N1s signals from N–H of TBA^+^ at 401.1 eV and P=N at 398.8 eV, as well as the O1s signals from C–O–P at 533.8 eV and V–O at 531.6 eV. Moreover, the P2p signals at 135, 134, and 132.9 eV (Figure 2C) can be assigned to P–Cl (I), P=N (II), and C–O–P (III), respectively, indicating that most of the phosphazenes were involved in the cross-linked framework [14,39,40]. Furthermore, as shown in Figure 2D, the signals of V2p at 524 and 516.7 eV indicated that the valence state of V mostly retained +5, demonstrating that their redox properties were retained in the resulted HCCP-V, and enabled their potential application in the selective oxidation of sulfides.

As expected, the resulted HCCP-V displayed a spherical aggregated structure with a diameter of ca. 60 nm (Figure 3A and Appendix A). Due to the homogeneous distribution of the anionic [V_6_O_13_]^2−^ cluster in the cyclomatrix structure, the resulted HCCP-V displayed a negative zeta potential of −32 mV (Figure 3B), which also indicated the relatively stable nature of such particles. In agreement with our previous study, the rigidity of the V_6_O_13_-PhOH cluster supported the porous structure of the resulted HCCP-V very well (typical IV-type isotherm), although the BET surface area was as low as 12.73 m^2^/g (Figure 3C) due to the possible occupation of the pores by the bulky TBA cations [34,41]. The pore size distribution calculated from the desorption curve mainly ranged from 30 to 65 nm (Figure 3D), revealing the nature of the mesoporous structure.

### 2.2. Catalytic Oxidation of MPS by HCCP-V

The [V_6_O_13_]^2−^ clusters are known to be active towards the oxidation of sulfides [42]. To demonstrate the applicability of HCCP-V for H_2_O_2_-based oxidative removal reactions, different sulfides, including MPS, DBT, and CEES, were used as substrates in the catalytic oxidation [43] (as shown in Figure 2).

Firstly, the catalytic oxidation of MPS was used as a model reaction to explore the heterogeneous catalytic activity of HCCP-V. Generally, the catalytic oxidation was carried out at different temperatures (25 °C, 40 °C, and 55 °C) with [MPS]:[H_2_O_2_]:[catalyst] = 1:1.2:1/400. As a control experiment, it can be found that no oxidation was observed when no catalyst was added or organic cyclomatrix phosphazenes HCCP-BPS was used (Appendix A). In contrast, the presence of V_6_O_13_-PhOH in HCCP-V enables the catalytic oxidation of MPS to methyl phenyl sulfoxide (MPSO). The reaction was monitored by HPLC (Figure 4), and it can be seen that with the prolongation of the reaction time, the peak of MPS at 7.6 min gradually decreased, and the peak of MPSO at 2.7 min increased, which is in good agreement with corresponding ^1^H NMR results (Appendix A). More importantly, the overoxidation was relatively suppressed, as less than 2% of methyl phenyl sulfone (MPSO2) was detected at a retention time of 3.3 min. The oxidation of MPS can be completed within 180 min with MPSO conversion as high as 99% at 25 °C. Moreover, the reaction rate can be promoted at high temperatures. For example, the reaction can be completed within 20 min at 55 °C (Table 1). The relationship between ln(*C*_t_/*C*_0_) and the reaction time reveals that the kinetics of the catalytic oxidation of MPS by HCCP-V follows the second-order kinetics, with the highest reaction rate constant of 0.01189 min^−1^ at 55 °C.

By comparing the catalytic oxidation results at different temperatures (Figure 4 and Table 1) it can be found that the reaction rate was greatly improved with the increase of the reaction temperature. However, the conversion of MPS was reduced at high temperatures. Therefore, 40 °C was chosen as the optimized reaction temperature for catalytic oxidation of MPS.

To explore the effect of the oxidant dosage on the catalytic oxidation of MPS, the reaction was also investigated with [MPS]:[H_2_O_2_]:[catalyst] = 1:1:1/400 at different temperatures (25 °C, 40 °C, and 55 °C). As shown in Figure 5, the oxidation of MPS was completed in 120 min, and the conversion of MPS was 85.6% at 25 °C, which is lower than that of [MPS]:[H_2_O_2_]:[catalyst] = 1:1.2:1/400. Therefore, a slightly excess amount of the oxidant should be better for such kind of reaction. Similarly, increasing the reaction temperature can accelerate the oxidation by reducing the reaction time from 120 min at 25 °C to almost 12 min at 55 °C. However, the conversion of MPS was also decreased.

By comparing the catalytic results of MPS with different ratios of HCCP-V (Table 1), it can be concluded that (i) the reaction rate and TOF increased with the temperature, and (ii) under the same temperature, the excess amount of oxidant favored the catalytic oxidation reaction.

### 2.3. Catalytic Oxidation of DBT by HCCP-V

Besides MPS, dibenzothiophene (DBT) was also selected as the substrate to explore the potential application of HCCP-V in the oxidative desulfurization of diesel [44,45]. Generally, the oxidation of DBT was more challenging than MPS. Therefore, excess amounts of catalyst and high temperatures were usually used in the oxidation of DBT. The catalytic experiment was monitored by HPLC and performed at 70 °C (or 80 °C) in acetonitrile (solvent, 5 mL) with DBT (115.16 mg, 1 eq), catalyst (10 mg, 1/100 eq), H_2_O_2_ (313.4 μL, 5 eq or 0.5 mL, 8 eq), and naphthalene (internal standard). As shown in Figure 6A–C, in the case of 70 °C and 5 eq of H_2_O_2_, the peak of DBT with a retention time of 17.6 min gradually decreased, and the peaks of DBTSO at 3.7 min and DBTSO2 at 4.6 min gradually increased, indicating the successful catalytic oxidation of DBT. However, only 79.8% conversion of DBT and 52.5% conversion of DBTSO2 were achieved after 50 min, indicating that the amount of oxidant was insufficient to fully convert DBT. Moreover, the oxidative desulfurization catalyzed by HCCP-V followed the second-order kinetics with a reaction rate constant of 0.0007 min^−1^ (Figure 6C). By increasing the temperature to 80 °C, the reaction time was reduced from 50 min for 70 °C to 15 min, however, the conversion of both DBT and DBTSO2 was also decreased (Figure 6D–F).

Interestingly, by increasing the dosage of H_2_O_2_ from 5 to 8 eq and keeping the temperature at 70 °C, the oxidative desulfurization of DBT can be completed within 50 min. More importantly, the conversion curves of DBT and DBTSO2 were greatly improved to 92.2% and 98.9%, respectively (Figure 6G–I, Table 2).

### 2.4. Catalytic Oxidation of CEES by HCCP-V

To further explore the potential application of HCCP-V in the decontamination of chemical warfare agents, the catalytic oxidation of the mustard simulant 2-chloroethyl ethyl sulfide (CEES) [46,47] has been explored. As monitored by ^1^H NMR (Figure 7A), the proton *b* in CEES gradually disappeared and proton *f* in the oxidized product CEESO appeared in the down field [24], indicating that CEES is completely and rapidly oxidized in the presence of HCCP-V and H_2_O_2_ at room temperature, showing its promise as an effective catalyst for the removal of mustard under mild conditions. Moreover, this reaction selectively forms the less toxic 2-chloroethyl ethyl sulfoxide (CEESO) without overoxidation to the harmful 2-chloroethyl ethyl sulfone (CEESO2) product (Figure 7B).

### 2.5. Recyclability of HCCP-V

All the above results demonstrated the versatility of HCCP-V in the catalytic oxidation of sulfides. To investigate the stability of such heterogeneous catalyst, the recyclability of HCCP-V was studied. Generally, the model reaction of MPS catalytic oxidation was carried out with oxidant dosage of 1.2 eq and temperature of 40 °C. According to HPLC and the conversion curves (Appendix A), the reaction time for the complete oxidation of MPS was ca. 20 min, and the conversion of MPS was above 99% during 4 cycles, indicating the tight immobilization of V_6_O_13_-PhOH in the polymeric framework and the robustness of the catalyst. However, the conversion of MPSO in repeated experiments slightly decreased from 85.72% to 72.65% (Figure 8A). More importantly, the structure of the catalyst also remained intact, as shown in Figure 8B. It can be found that after 4 cycles of catalytic oxidation, the peaks of the V–O bond at 954 cm^−1^ and the V–O–V bond at 754 and 681 cm^−1^ did not change, indicating that the structures of V_6_O_13_-PhOH clusters were stable.

## 3. Materials and Methods

### 3.1. Materials

Hexachlorocyclotriphosphazene (HCCP) and sodium metavanadate were purchased from Aladdin Biochemical Technology Co., Ltd. (Shanghai, China). Pentaerythritol and *p*-aminophenol were supplied by Shanghai Macklin Biochemical Co., Ltd. (Shanghai, China). HCCP was purified by sublimation in vacuum at 60 °C, three times. Acetonitrile, dimethyl sulfoxide (DMSO), and *N*,*N*-dimethylformamide (DMF) were stirred overnight with CaH_2_ and distilled before use. Triethylamine (TEA) was dried with KOH and distilled before use. Other reagents and chemicals were analytical grade and used as received.

### 3.2. Synthetic Procedures of Polyoxovanadate-Based Cyclomatrix Polyphosphazene Microspheres

#### 3.2.1. Synthesis of [N(C_4_H_9_)_4_]_2_[V_6_O_13_{(OCH_2_)_3_CCH_2_OH}_2_] (V_6_O_13_-OH)

The synthetic procedure of V_6_O_13_-OH has been reported already [48], and the modified method was as follows: NaVO_3_ (4 g, 2 eq) and pentaerythritol (2.23 g, 1 eq) were dissolved in 50 mL of deionized water at 60 °C. After cooling to room temperature, the pH of the solution was adjusted to 1.0 with 1.0 M HCl. The reaction mixture was heated at 80 °C for 6 h in the dark. Then, the dark green insoluble precipitate was removed by filtration to afford a deep red solution. To this solution, tetrabutylammonium bromide (TBABr) aqueous solution (4 g in 25 mL) was added dropwise and stirred for 1–2 h at room temperature. The resulted brick red precipitate was collected by filtration and washed with ethanol 3 times to afford the final product. Yield: 34%, based on V. FT-IR (KBr, cm^−1^): 3410 (−OH, m), 2961 (CH, s), 2923 (CH, s), 2853 (CH, m), 1637 (−OH, w), 1480 (CH, s), 1380 (s), 1126 (s), 1130 (m), 1067 (m), 1039 (C−O, m), 956 (V−O, s), 944 (vs), 811 (V−O−V, m), 720 (V−O−V, s), 582 (m). ^1^H NMR (400 MHz, DMSO-*d_6_*, *δ*): 5.74 (s, 2H, −OH), 4.87 (s, 12H, −CH_2_C−), 4.46 (s, 4H, −CCH_2_−), 3.15 (br, 16H, −NCH_2_−), 1.56 (br, 16H, −CH_2_−), 1.30 (br, 16H, −CH_2_−), 0.93 (br, 24H, −CH_3_).

#### 3.2.2. Synthesis of [N(C_4_H_9_)_4_]_2_[V_6_O_13_{(OCH_2_)_3_CCH_2_OCONHC_6_H_4_OH}_2_] (V_6_O_13_-PhOH)

The solutions of *p*-aminophenol (523.8 mg, 1 eq) in 4 mL of DMSO and *N*,*N’*-carbonyldiimidazole (CDI, 934 mg, 1.2 eq) in 2 mL of DMSO were degassed by purging N_2_ for 30 min. The CDI solution was added dropwise to the Schlenk flask with *p*-aminophenol under N_2_ and stirred at room temperature in the dark for 5 h to obtain a 0.8 M stock solution. V_6_O_13_-OH (1.64 g, 1 eq) was dissolved in 7 mL of dry acetonitrile and purged N_2_ for 30 min in the dark. Then, 3.9 mL of the stock solution was added dropwise to the above solution. Dibutyltin dilaurate (0.92 mL, 1.2 eq) was used as a catalyst, and the reaction was stirred at 80 °C in the dark for 60 h. The reaction was monitored with FT-IR. After the reaction, precipitates were removed by centrifugation (9000 rpm, 5 min), and the supernatant was concentrated and added dropwise to TBABr aqueous solution (4 g in 25 mL). The resulted precipitate was collected and washed with dichloromethane and deionized water to afford the target compound. Yield: 44%, based on V. FT-IR (KBr, cm^−1^): 3280 (−NH, m), 2957 (CH, s), 2925 (CH, s), 2871 (CH, m), 1744 (w), 1693 (C=O, m), 1606 (Ph, s), 1553 (Ph, w), 1460 (CH, s), 1378 (s), 1220 (s), 1130 (m), 1068 (m), 1032 (C−O, m), 951 (V−O, s), 944 (vs), 835 (V−O−V, m), 712 (V−O−V, s), 579 (m) [49]. ^1^H NMR (DMSO-*d_6_*, 400 MHz, *δ*): 9.04 (s, 2H, −OH from phenol), 8.19 (s, 2H, −NH−), 7.17 (br, 4H, Ar−H), 6.67 (br, 4H, Ar−H), 4.87 (s, 6H, −CH_2_C−), 4.47(s, 4H, −CCH_2_−), 3.18 (br, 16H, −NCH_2_−), 1.57 (br, 16H, −CH_2_−), 1.23 (br, 16H, −CH_2_−), 0.93 (br, 24H, −CH_3_).

#### 3.2.3. Precipitation Polymerization to Prepare Polyoxovanadate-Based Cyclomatrix Polyphosphazene Microspheres (HCCP-V)

The preparation was similar to our previous method [14,50], and the detailed procedure was as follows: V_6_O_13_-PhOH (0.8 g, 3 eq), TBABr (1.12 g, 20 eq), and HCCP (60.26 mg, 1 eq) were dissolved in a mixture solvent of 0.2 mL of DMF and 2.5 mL of acetonitrile. The reaction mixture was degassed with N_2_ for 30 min, followed by the addition of TEA (0.44 mL, 18 eq), then stirred at 90 °C in the dark for 72 h. The reaction was monitored with FT-IR. After the reaction, the resultant precipitate was collected by centrifugation (9000 rpm, 5 min), then washed with acetonitrile and ethanol to afford the target product. Yield: 71%. FT-IR (KBr, cm^−1^): 3140 (−NH, m), 2920 (CH, s), 2852 (CH, m), 1606 (Ph, s), 1543 (N−H, s), 1508 (Ph, m), 1238 (P=N, s), 955 (V−O, s), 906 (P−O−Ph, s), 805 (V−O−V, m), 706 (V−O−V, s), 650 (s).

### 3.3. Characterization

FT-IR spectra were recorded on a Bruker TENSOR27 with a resolution of 0.4 cm^−1^ over the range of 4000−400 cm^−1^. ^1^H NMR spectra were recorded with a Bruker Avance 400 spectrometer at 400 MHz in CDCl_3_ and DMSO-*d_6_* using tetramethylsilane (TMS) as an internal standard. Thermal stability was investigated on a Mettler Toledo TGA 2 instrument with a heating rate of 10 °C/min in O_2_ atmosphere. X-ray photoelectron spectroscopy (XPS) spectra were recorded on a Kratos AXIS Ultra DLD spectrometer with a monochromatic Al Kα X-ray source. The nitrogen adsorption and desorption isotherm was measured at 77 K on an American Mike TriStar II 3020 analyzer. The morphology of the sample was observed on a FEI Verios G4 scanning electron microscope (SEM). The samples were coated with a thin sputtered Au before SEM characterization. High-performance liquid chromatography (HPLC) results were collected by the Shimadzu Essentia LC-16 with ultraviolet detector.

### 3.4. The Catalytic Oxidation Experiments

General procedure of the catalytic oxidation of methyl phenyl sulfide (MPS): MPS (78 mg, 1 eq) was dissolved in 5 mL of acetonitrile, and naphthalene (80.1 mg, 1 eq) was added as an internal standard. The mixture was stirred for 10 min (500 rpm) and an aliquot was taken as the t_0_ sample. Freshly ground HCCP-V (2.5 mg, 1/400 eq) was added as a catalyst, and after stirring for 10 min, 30 wt.% H_2_O_2_ (77 μL, 1.2 eq or 63 μL, 1 eq) was added as an oxidant. The reaction temperature was set at 25 °C, 40 °C, or 55 °C, respectively, and the whole process was monitored by HPLC (1 mL/min, acetonitrile:H_2_O = 7:3, injection volume: 10 μL, detector wavelength: 254 nm). After the reaction, the catalyst was collected by centrifugation, and the final product was characterized by ^1^H NMR.

General procedure of the catalytic oxidation of dibenzothiophene (DBT): DBT (115.16 mg, 1 eq) was dissolved in 5 mL of acetonitrile, and naphthalene (80.1 mg, 1 eq) was added as an internal standard. The mixture was stirred for 10 min (500 rpm) and an aliquot was taken as the t_0_ sample. Freshly ground HCCP-V (10 mg, 1/100 eq) was added as a catalyst, and after stirring for 10 min, 30 wt.% H_2_O_2_ (313.4 μL, 5 eq or 0.5 mL, 8 eq) was added as an oxidant. The reaction temperature was set at 70 °C or 80 °C, and the whole process was monitored by HPLC (the condition was the same as the one for MPS).

General procedure of the catalytic oxidation of 2-chloroethyl ethyl sulfide (CEES): CEES (75 μL, 1 eq) was dissolved in 5 mL of acetonitrile. After stirring for 10 min (500 rpm), an aliquot was taken as the t_0_ sample. Freshly ground HCCP-V (2.5 mg, 1/400 eq) was added as a catalyst, and after stirring for 10 min, 30 wt.% H_2_O_2_ (77 μL, 1.2 eq) was added as an oxidant. The whole process was monitored by ^1^H NMR.

## 4. Conclusions

Functional polyoxovanadate [V_6_O_13_]^2−^ were successfully immobilized to cyclomatrix polyphosphazene microspheres via precipitation polymerization. The rigidity and anionic nature of [V_6_O_13_]^2−^ endowed the resulted HCCP-V with porosity and a negative charged surface. Owing to the homogeneous distribution of the [V_6_O_13_]^2−^ cluster in the network and the high stability of the P=N framework, HCCP-V exhibited versatility in the catalytic oxidation of sulfides. By using H_2_O_2_ as an oxidant, the conversion of MPS, DBT, and CEES could be achieved as high as 99.6%, 92.2%, and 100% within 60 min, respectively. Moreover, the selectivity of MPSO, DBTSO2, and CEESO could be as high as 99%. Furthermore, the conversion of MPS was above 99% during 4 cycles, demonstrating the high stability and recyclability of such heterogeneous catalyst. This work provides facile methodology for the preparation of POMs-based MCPs for selective oxidation of sulfides. Moreover, the catalytic oxidation activity may be further improved by introducing other functional groups to the cyclomatrix polyphosphazene structure via synergetic interaction.

## Data Availability

Not applicable.

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
