# Peer review of "Polyoxovanadate-Based Cyclomatrix Polyphosphazene Microspheres as Efficient Heterogeneous Catalysts for the Selective Oxidation and Desulfurization of Sulfides"

_molecules, 2022, doi:10.3390/molecules27238560_

Round 1

Reviewer 1 Report

Dear Ms. Hanna Xiong,

The manuscript (molecules-2038994) entitled ‘’ Polyoxovanadate-based Cyclomatrix Polyphosphazene Microspheres as Efficient Heterogeneous Catalysts for the Selective Oxidation and Desulfurization of Sulfides’’ includes the facile precipitation polymerization between phenol group modified [V6O13]2− and hexachlorocyclotriphosphazene and their characterizations. This article can be published in the Molecules journal after moderate revision.

My suggestions:

Line 72, Draw the open structure of the CDI (N, N'-carbonyldiimidazole) in C for easier understanding of the reactions and the drawing in B. Like

Line 265, the pH of the solution was adjusted to 1 with 1 M HCl.? It is better to write that the pH of the solution was adjusted to 1.0 with 1.0 M HCl.

Line 284, Ditin butyl dilaurate should be corrected as Dibutyltin dilaurate

Line 290, 1744 (−COO−, w), 1693 (C=O, m), 1378 (s), it should be two peaks for asym COO (1693 cm-1) and sym COO (1378 cm-1). Please delete the third one. I suggest one reference at this step: Applied Catalysis A: General, 433, 223-228, 2012.

Line 292, 9.04 (s, 2H, −OH)?? These kinds of OH protons appear at around 5.35 ppm.

Lines 323, 332, and 338, ….taken as t0 sample. it should be corrected as …. taken as to sample (after t, not zero should be o).

Line 354, in conclusion section, ….via synergetic interaction[44,45]. Use the references in the appropriate section before the conclusion.

Scheme 1, Xiao and Wu at al. 2020, Which reference in the reference section? 30. reference? If the reference is 30, there is an abbreviation and year error (Huang et al., 2021), or was it forgotten to be written in the reference part?

In the text, some characteristic proton 1H-NMR data should be given and interpreted. It should be emphasized that these data are compatible with the literature.

Author Response

Reply to Reviewer 1:

Line 72, Draw the open structure of the CDI (N, N'-carbonyldiimidazole) in C for easier understanding of the reactions and the drawing in B. Like

A: the chemical structure of CDI was drawn in Scheme 1B.

Line 265, the pH of the solution was adjusted to 1 with 1 M HCl.? It is better to write that the pH of the solution was adjusted to 1.0 with 1.0 M HCl.

A: It was changed to “the pH of the solution was adjusted to 1.0 with 1.0 M HC” according to your suggestion.

Line 284, Ditin butyl dilaurate should be corrected as Dibutyltin dilaurate

A: Correction was made according to your suggestion.

Line 290, 1744 (−COO−, w), 1693 (C=O, m), 1378 (s), it should be two peaks for asym COO (1693 cm-1) and sym COO (1378 cm-1). Please delete the third one. I suggest one reference at this step: Applied Catalysis A: General, 433, 223-228, 2012.

A: Correction was made according to your suggestion and the reference was updated.

Line 292, 9.04 (s, 2H, −OH)?? These kinds of OH protons appear at around 5.35 ppm.

A: The peak at 9.04 is attributed to the –OH group from the phenol groups, we corrected it.

Lines 323, 332, and 338, ….taken as t0 sample. it should be corrected as …. taken as to sample (after t, not zero should be o).

A: Since the sample at the beginning of the reaction will be used in the calculation of conversion, we took the sample at the beginning of the catalysis and labled as t0 sample.

Line 354, in conclusion section, ….via synergetic interaction[44,45]. Use the references in the appropriate section before the conclusion.

A: Correction was made according to your suggestion.

Scheme 1, Xiao and Wu at al. 2020, Which reference in the reference section? 30. reference? If the reference is 30, there is an abbreviation and year error (Huang et al., 2021), or was it forgotten to be written in the reference part?

A: It is the same reference, we corrected the year in Scheme 1 to 2021.

In the text, some characteristic proton 1H-NMR data should be given and interpreted. It should be emphasized that these data are compatible with the literature.

A: The characteristic peaks of the V6O13-PhOH as well as the spectrum in monitor of the oxidation process were discussed and corresponding reference was updated.

Reviewer 2 Report

The manuscript is written in good language and is devoted to the actual topic of the study. The synthesis of composites based on polymers and metal complexes is an interesting area of ​​research both from the point of view of theoretical chemistry and  their further application.

Remarks:

1. The introduction should be expanded by adding information about vanadium polyoxometalates as compounds with unique properties.

2. Figure 3. It makes sense to calculate and present the particle size distribution based on microscopy data.

3. The accuracy of determining the specific surface area is excessive.

4. The pore size distribution curve needs to be smoothed out.

It is necessary to speciy from which curve (adsorption or desorption) the pore size distribution is calculated.

5. To obtain more information about the porous structure of materials, it is necessary to use the Brunauer and De Boer classifications to describe the adsorption isotherm.

Author Response

1. The introduction should be expanded by adding information about vanadium polyoxometalates as compounds with unique properties.

A: A new paragraph was added to introduce some unique properties of polyoxovanadates in the Introduction section.

2. Figure 3. It makes sense to calculate and present the particle size distribution based on microscopy data.

A: Particle size distribution was provided based on the statistic analysis of the SEM result, the Figure was added as an inset to Figure 3A.

3. The accuracy of determining the specific surface area is excessive.

A: the specific surface area was corrected to 12.73 m2/g.

4. The pore size distribution curve needs to be smoothed out.

It is necessary to speciy from which curve (adsorption or desorption) the pore size distribution is calculated.

A: the pore size distribution curve was smoothed according to your suggestion. The pore size distribution was calculated from the adsorption curve.

5. To obtain more information about the porous structure of materials, it is necessary to use the Brunauer and De Boer classifications to describe the adsorption isotherm.

A: according to Brunauer and De Boer classifications, the adsorption isotherm belongs to typical IV-type isotherm.

Round 2

Reviewer 2 Report

The authors listened to the remarks made and, on the whole, made the necessary corrections to the manuscript.

There is one more note that needs to be corrected:

The pore size distribution (Fig. 3) must be calculated from the desorption curve. Since in your case the materials have a slit form of pores. This can be seen from the shape of the hysteresis loop. The calculation from the adsorption curve is a methodological error.

After correction - the article can be published in the journal.

Author Response

Dear reviewer,

Thanks very much for your kind suggestion.  We double checked the original BET data, the pore size distribution in Fig. 3 was calculated from the desorption curve, corresponding correction was made in the revised manuscript.

Thanks 

Yi